# Transmission genetics of drug-resistant hepatitis C virus

Nicholas van Buuren[1], Timothy L Tellinghuisen[2], Christopher D Richardson[3], Karla Kirkegaard[1]*

[1]Department of Genetics, Stanford University School of Medicine, Stanford, United States; [2]Department of Infectious Diseases, The Scripps Research Institute, Jupiter, United States; [3]Department of Microbiology and Immunology, Dalhousie University, Nova Scotia, Canada

**Abstract** Antiviral development is plagued by drug resistance and genetic barriers to resistance are needed. For HIV and hepatitis C virus (HCV), combination therapy has proved life-saving. The targets of direct-acting antivirals for HCV infection are NS3/4A protease, NS5A phosphoprotein and NS5B polymerase. Differential visualization of drug-resistant and -susceptible RNA genomes within cells revealed that resistant variants of NS3/4A protease and NS5A phosphoprotein are *cis*-dominant, ensuring their direct selection from complex environments. Confocal microscopy revealed that RNA replication complexes are genome-specific, rationalizing the non-interaction of wild-type and variant products. No HCV antivirals yet display the dominance of drug susceptibility shown for capsid proteins of other viruses. However, effective inhibitors of HCV polymerase exact such high fitness costs for drug resistance that stable genome selection is not observed. Barriers to drug resistance vary with target biochemistry and detailed analysis of these barriers should lead to the use of fewer drugs.

DOI: https://doi.org/10.7554/eLife.32579.001

*For correspondence:
karlak@stanford.edu

## Introduction

In a recent triumph of modern science and medicine, patients chronically infected with hepatitis C virus (HCV) now receive multidrug regimens that are often curative and have low toxicity (*Lawitz et al., 2013*; *Afdhal et al., 2014*; *Dhaliwal and Nampoothiri, 2014*). Over the past two decades, researchers have developed and tested thousands of antiviral compounds with varying efficacies and toxicity profiles that have ultimately lead to the FDA approval of powerful combination therapies (*Lawitz et al., 2013*; *Scheel and Rice, 2013*). Several different direct-acting antivirals (DAAs) that target the NS3/4A protease, NS5A phosphoprotein, or NS5B RNA-dependent RNA polymerase of HCV have been approved for use in the clinic (*Afdhal et al., 2014*; *Dhaliwal and Nampoothiri, 2014*; *Manns and von Hahn, 2013*; *Younossi et al., 2015*). Ideally, the knowledge gained in developing HCV antivirals that are effective and not prone to the outgrowth of drug resistance will be applied to other viruses as well.

The emergence of drug-resistant variants follows basic evolutionary principles, requiring spontaneous mutations as well as selective pressure, so that beneficial mutations increase the progeny size of genomes that bear them. The genetic diversity in RNA viral genomes results from the high error frequencies incurred by RNA-dependent RNA polymerases, which occur at approximately $4 \times 10^{-5}$ errors for each nucleotide synthesized (*Sanjuán et al., 2010*). Given the iterative copying of positive and negative strands, much higher cumulative error frequencies are observed, even during a single cycle of infection (*Sanjuán et al., 2010*; *Acevedo et al., 2014*). When more than one mutation is required to confer drug resistance, the outgrowth of drug resistance can be delayed (*Bloom et al., 2010*). As a result, treatment with combinations of drugs can be extremely effective at suppressing

**eLife digest** Viruses are simple organisms that consist of genetic information and a few types of proteins. They cannot replicate on their own, and instead hijack the molecular machinery of a host cell to produce more of themselves.

Inside an infected cell, the genetic information of the virus is replicated and 'read' to create viral proteins. These components are then assembled to form a new generation of viruses. During this process, genetic errors may occur that lead to modifications in the viral proteins, and help the virus become resistant to treatment. For instance, a viral protein that used to be targeted by a drug can change slightly and not be recognized anymore.

Currently, the most efficient way to fight drug resistance is to use combination therapy, where several drugs are given at the same time. This strategy is successful, for example to treat infections with the hepatitis C virus, but it is also expensive, especially for developing countries.

An alternative approach is dominant-drug targeting, which exploits the fact that both drug-resistant and drug-susceptible viruses are 'born' in the same cell. There, the susceptible viruses can overwhelm and 'mask' the benefits of the resistant ones. For example, proteins from resistant strains, which are no longer detected by a treatment, can bind to proteins from susceptible viruses; drugs will still be able to recognize these resulting viral structures. The proteins that operate in such ways are potential dominant-drug targets. However, resistant and susceptible strains can also cohabit without any contacts if their proteins do not interact with each other.

Now, van Buuren et al. screen several viral proteins, including one called NS5A, to test whether a dominant drug target exists for the hepatitis C virus. Only a few molecules of a drug that targets NS5A can stop the virus from growing. In theory, drug-bound NS5A proteins could block their non-drug-bound neighbors, but when these drugs have been used on their own, resistance quickly emerged.

Experiments showed that NS5A is not a dominant drug target because the drug-resistant and drug-susceptible proteins do not mix. Unless 'forced' in the laboratory, NS5A proteins only bind to the ones produced by the same strain of virus. This explains why resistant viruses quickly take over when NS5A drugs are the sole treatment. However, other hepatitis C proteins, such as the HCV core protein, are known to mix during the assembly of the virus, and thus are likely be dominant drug targets.

DOI: https://doi.org/10.7554/eLife.32579.002

drug resistance, because the number of mutations required for resistance to multiple drugs is ideally the sum of the number of mutations needed for each drug alone. Combination therapies have proven invaluable in reducing the frequency of drug resistance in both microbiology and oncology (*Fillat et al., 2014*; *Falade-Nwulia et al., 2017*; *Kerantzas and Jacobs, 2017*).

Other strategies to suppress viral drug resistance accept the inevitability of drug-resistant mutations, but seek to decrease selection for their outgrowth. Examples of antivirals for which resistance comes with a high fitness cost include integrase inhibitors of HIV (*Mesplède et al., 2015*), protease inhibitors of coronaviruses (*Deng et al., 2014*) and certain nucleoside inhibitors of HCV NS5B polymerase (*Lawitz et al., 2013*). As was first shown for 2'-C-methyl CTP, selected drug-resistant HCV variants grow poorly and retain their low fitness upon passage (*Dutartre et al., 2006*). Sofosbuvir, the FDA-approved NS5B polymerase inhibitor, has dramatically increased the efficacy of HCV treatment, and also generates little outgrowth of resistant variants. The few HCV variants observed in patients are nearly inviable (*Svarovskaia et al., 2016*). Understanding the mechanisms by which this kind of fitness cost is enforced would greatly facilitate future antiviral design.

Another approach to decrease the selection of drug-resistant variants is termed dominant drug targeting. This applies to antiviral targets for which the drug-bound products of pre-existing drug-susceptible genomes are dominant-negative inhibitors of new drug-resistant progeny (*Crowder and Kirkegaard, 2005*; *Tanner et al., 2014*; *Mateo et al., 2015*). Recently, this has been demonstrated for the capsid proteins of poliovirus and dengue virus (*Tanner et al., 2014*; *Mateo et al., 2015*), but other potential dominant drug targets have also been identified (*Crowder and Kirkegaard, 2004*). When a drug-resistant genome is in its cell of origin, it coexists with its drug-susceptible parents and

siblings. If the drug target is, for example, a subunit of an oligomeric complex and subunits from different genomes have the opportunity to mix, chimeric oligomers often form. At the time of its creation, the drug-resistant genome will be a minority species, and such chimeras would be predominantly composed of the drug-bound, susceptible components thus incapacitating the entire oligomeric structure. Such 'phenotypic masking' was originally invoked to explain the very low frequency of foot-and-mouth-disease escape variants following selection with neutralizing antibodies when passaged at high multiplicities of infection (MOIs)(*Holland et al., 1989*).

Our goal was to screen the HCV-encoded viral proteins that are current targets of antiviral compounds to determine the intracellular dominance relationships between drug-resistant and drug-susceptible genomes. The high cost to viral fitness of Sofosbuvir-resistant variants is sufficient to explain its high barrier to resistance. There are currently no antivirals directed against HCV core protein; however, it is likely to be a dominant drug target. We used differential hybridization of RNA probes to detect two different genomic RNAs in a single cell by confocal microscopy and by flow cytometry. This analysis showed the *cis*-dominance of HCV viruses that are resistant to inhibitors of either NS3/4A protease or NS5A phosphoprotein, consistent with the rapid outgrowth of drug-resistance in patients of these two inhibitor classes.

## Results

### Construction of three strains of codon-altered JFH1

Newly mutated drug-resistant genomes first arise within cells that are pre-populated by drug-susceptible genomes. To mimic such mixed infections, we have previously employed co-infection of cultured cells with drug-susceptible and drug-resistant viruses at high MOIs to ensure mixed infection (*Tanner et al., 2014*; *Mateo et al., 2015*). For HCV, it is not practical to use high MOIs to achieve co-infection due to the difficulty of obtaining sufficiently high-titer viral stocks. Thus, we needed to develop an approach to distinguish between uninfected, singly infected and co-infected cells in relatively sparsely infected cell populations (*Figure 1A*).

To detect individual genomes in infected cells, a single-molecule fluorescence in situ hybridization (FISH) approach was used. A recently developed branched DNA probe technology allows the generation of sufficiently sensitive RNA probes to identify single molecules within cells, but requires approximately 1000 nucleotides of differential probe hybridization to achieve specificity (*Affymetrix Inc, 2016*). To create a viral strain with this extreme dissimilarity from wild-type virus, we tested the viability of three different codon-altered versions of the JFH1 variant of HCV (*Figure 1B*). Each mutated version contained 200–300 nucleotide changes that did not alter the protein sequence (*Figure 1—figure supplements 1–3*). Of these codon-altered (CA) variants, CA-1 was inviable, CA-2 showed reduced viral protein accumulation, and CA-3 showed accumulation of both viral protein and RNA to abundances equivalent to those of the wild-type virus (*Figure 1C*). Recently, detailed analysis of covarying nucleotides within the HCV coding region has identified the location of several previously unknown functional RNA secondary structures (*Pirakitikulr et al., 2016*). Interestingly, CA-1 contains two such regions and CA-2 contains one, which correlates with decreasing viability, while CA-3 contains no such regions (*Figure 1—figure supplements 1–3*) (*Pirakitikulr et al., 2016*). Thus, subsequent experiments were performed only with CA-3. This variant, now termed 'CA' virus, contains 247 synonymous mutations over a 918-nucleotide region that spans the coding sequences for most of NS2 and the N-terminus of NS3 (*Figure 1—figure supplement 3*).

To test the sensitivity of RNA FISH probes generated against the positive- and negative-strands of wild-type (WT) and codon-altered (CA) viruses, both confocal microscopy and flow cytometry analyses were employed. Branched DNA technology allowed the labeling of each target RNA with as many as 8000 fluorophores (*Figure 2A*)(*Affymetrix Inc, 2016*). Huh-7.5.1 cells were infected with either WT or CA viruses, subjected to FISH and visualized by confocal microscopy. WT and CA probe sets specifically targeted either the positive-sense (*Figure 2B*) or the negative-sense vRNA (*Figure 2C*) of their corresponding virus. Additionally, we tested whether flow cytometry efficiently resolved cells transfected with different vRNAs; transfection was used to maximize the yield of each population. We resolved cells transfected with WT vRNA (*Figure 2Di*), transfected with CA vRNA (ii), a mixture of these two cell types (iii) and cells co-transfected with both WT and CA vRNAs (iv). Thus,

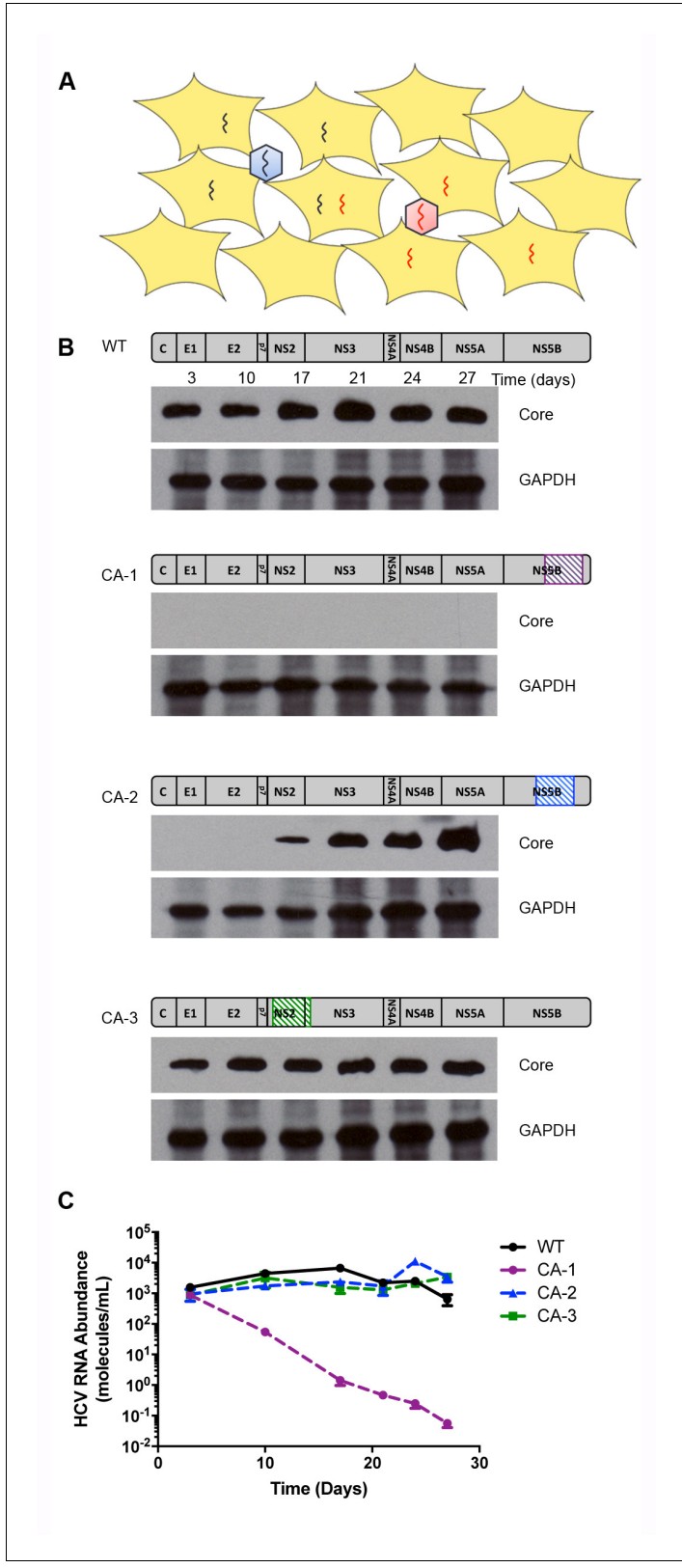

**Figure 1.** Construction of codon altered JFH1. (**A**) Cell cultures coinfected with two strains of HCV result in four populations: uninfected, two types of singly infected, and coinfected cells. (**B**) Three segments of the JFH1 genome, that were roughly 1000 nucleotides in length and had altered codon usage, were designed using GeneArt algorithms and synthesized. These genome fragments were then cloned into the JFH1 strain of HCV.

*Figure 1 continued on next page*

*Figure 1 continued*

Huh7.5.1 cells were transfected with each construct using electroporation to create long-term HCVcc cultures and viability was followed over time by immunoblotting of cell lysates for the HCV core protein. (C) Viral passages shown in (B) were monitored by qRT-PCR analysis of viral RNA in culture supernatants. Only the CA-3 virus demonstrated growth kinetics comparable to wild-type (WT) JFH1.

DOI: https://doi.org/10.7554/eLife.32579.003

The following figure supplements are available for figure 1:

**Figure supplement 1.** Alignment of codon-altered and wild-type JFH1 viral RNAs.
DOI: https://doi.org/10.7554/eLife.32579.004
**Figure supplement 2.** Alignment of codon-altered and wild-type JFH1 viral RNAs.
DOI: https://doi.org/10.7554/eLife.32579.005
**Figure supplement 3.** Alignment of codon-altered and wild-type JFH1 viral RNAs.
DOI: https://doi.org/10.7554/eLife.32579.006

specific RNA probes could be used to monitor the fate of drug-susceptible and drug-resistant viruses in co-infected cells.

## Transmission genetics and phenotypic dominance of drug-resistant NS3 variant D168A

To test the genetic properties of viruses that are resistant to NS3/4A inhibitors, we employed the original NS3/4A inhibitor, BILN-2061 (*Figure 3A*)(*Lamarre et al., 2003*). Like other NS3/4A inhibitors, BILN-2061 treatment rapidly allows the selection of drug resistant variants both in tissue culture and in patients (*Lamarre et al., 2003*; *Lin et al., 2004*). Given the ease of outgrowth of drug-resistant variants, we hypothesized that NS3/4A was not a dominant drug target and that drug resistance would be genetically dominant. NS3-D168A is the prototypic mutation associated with resistance to NS3/4A inhibitors. Asp168 is in close proximity to the protease active site (*Figure 3B*). The ability of the NS3-D168A mutation to confer resistance to BILN-2061 was confirmed in both the WT and CA backgrounds (*Figure 3—figure supplement 1*).

As shown schematically in *Figure 3D*, the ability to track cells that are uninfected (U), singly infected with drug-susceptible virus (S), infected with both susceptible and resistant virus (S + R) and singly infected with drug-resistant virus (R), can reveal dominance relationships during co-infection. In the absence of a drug, all viral populations should be present. However, in the presence of a drug, three outcomes are possible depending on the genetic outcome within the R + S population. If drug resistance were *trans*-dominant (*Figure 3E*), the drug-resistant virus would rescue the drug-susceptible genomes and all viruses in R + S cells would survive in the presence of the drug. S cells would drop into the U population, and R cells would survive. If drug resistance were *cis*-dominant (*Figure 3F*), only the R viruses in the R + S cells would survive, because the drug-resistant proteins would be unable to rescue the S viruses in the same cell. Consequently, the R + S cells would drop into the R population. If drug susceptibility were dominant (*Figure 3G*), all viruses in the R + S cells would be cleared, and the R + S cells, like the S cells, would drop into the U population, and only the R cells would continue to replicate.

To determine the dominance relationship between BILN-2061-susceptible and the BILN-2061-resistant virus, Huh-7.5.1 cells were infected with CA and WT-D168A viruses (*Figure 3C*). Cells were infected for 72 hr at MOIs such that all four populations were represented, followed by 36 hr of continued incubation in the absence or presence of 2 μM BILN-2061. Cells were then harvested, fixed, co-stained with wild-type and codon-altered RNA probe sets and analyzed by flow cytometry. All four cell types appeared in the absence of BILN-2061 (*Figure 3H,I*). In the BILN-2061-treated samples (*Figure 3J,K*), the susceptible S population shifted to the U cells as expected. The S + R cells, on the other hand, shifted to the R population upon drug treatment. Thus, the drug-resistant viral genomes in the co-infected cells could replicate, but could not rescue the drug-susceptible ones. Data from this and replicate experiments (*Figure 3I,K*) confirmed the quantitative shift of S + R cells into the R population upon drug treatment. We conclude that, for the NS3/4A target, drug-resistant genomes are *cis*-dominant for the 1:1 ratio of S and R viruses tested here. We also tested whether over-expressed drug-resistant NS3/4A precursors could rescue BILN-2061-susceptible virus (*Figure 3—figure supplement 2B*). Salvage of S virus was not observed. Importantly, when *cis*-acting

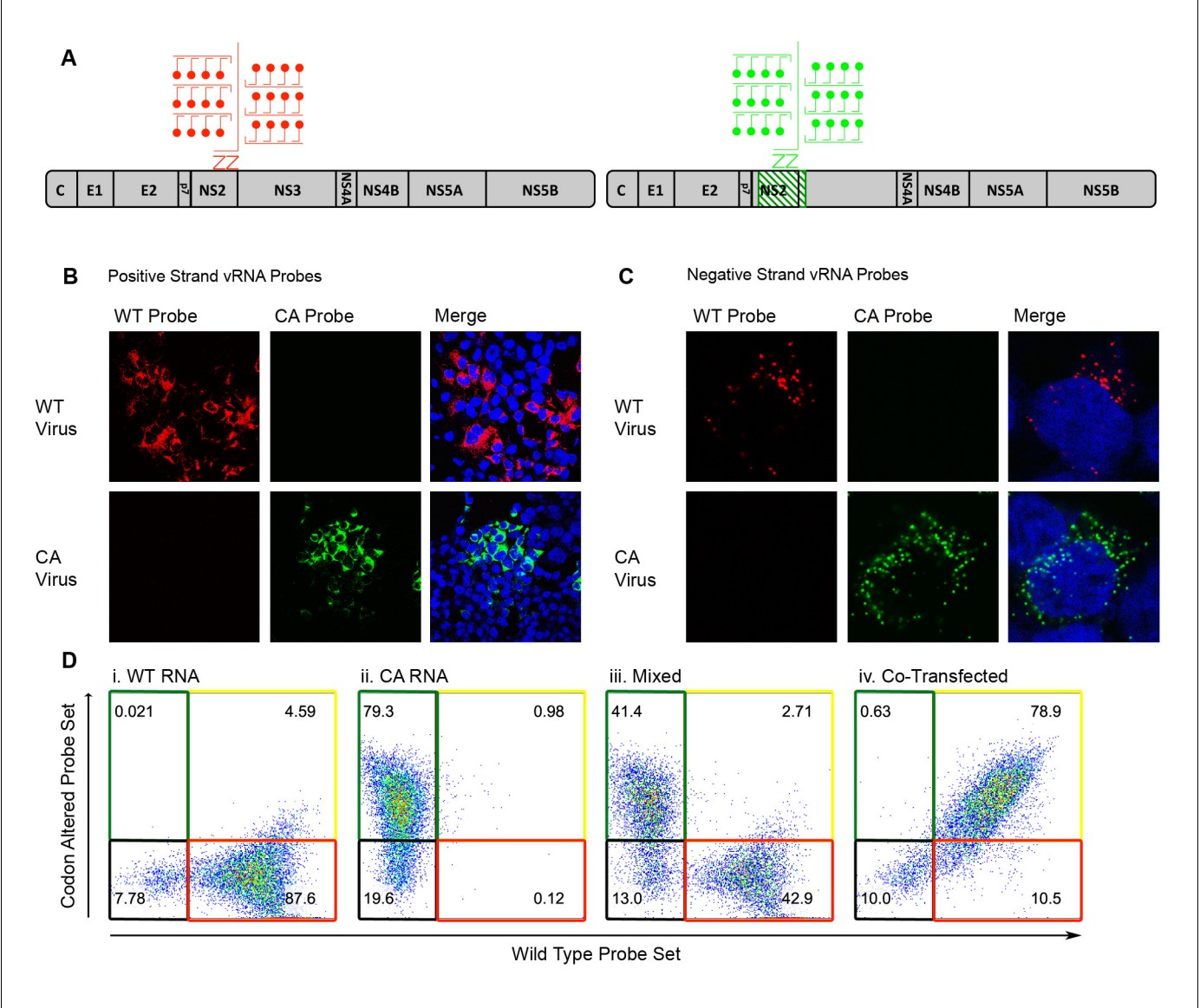

**Figure 2.** Differentiation of codon-altered and wild-type JFH1 using confocal microscopy and flow cytometry. (**A**) RNA in situ hybridization probes were designed to differentiate between wild-type (WT) and codon-altered (CA) viral RNAs. These probes utilize branched DNA technology to amplify the contiguous DNA branches and not the RNA target itself. Roughly 8000 fluorophores labeled each target RNA. (**B**) Huh7.5.1 cells on coverslips were infected with WT or CA virus at MOIs of 0.01 FFU/cell. Cells were fixed after 72 hr, co-stained with WT and CA probe sets that recognized HCV positive strands and visualized by confocal microscopy. (**C**) Confocal microscopy of cells infected as in (B) but co-stained with probe sets to identify negative strands. (**D**) Huh7.5.1 cells were transfected with WT RNA, CA RNA, or both by electroporation. Cells were fixed at 72 hr post transfection and costained with WT and CA RNA probe sets. Flow cytometry was performed on (i) cells transfected with WT RNA, (ii) cells transfected with CA RNA, (iii) a mixture of cells in i and ii, and (iv) cells transfected with both WT and CA RNAs.

DOI: https://doi.org/10.7554/eLife.32579.007

proteins are drug targets, drug-resistant products will enhance the propagation of only those genomes that encode them, allowing powerful selection for drug resistance.

## Fitness cost of resistance to NS5B inhibitor R1479

For NS5B polymerase inhibitor Sofosbuvir, the few resistant viral variants that arise in patients are highly attenuated. To investigate whether a related compound, R1479 (*Klumpp et al., 2006*), exacted a similar cost to viral fitness to drug-resistant variants, we attempted to recover R1479-resistant viruses for dominance testing. JFH1 was passaged for multiple rounds of infection in the

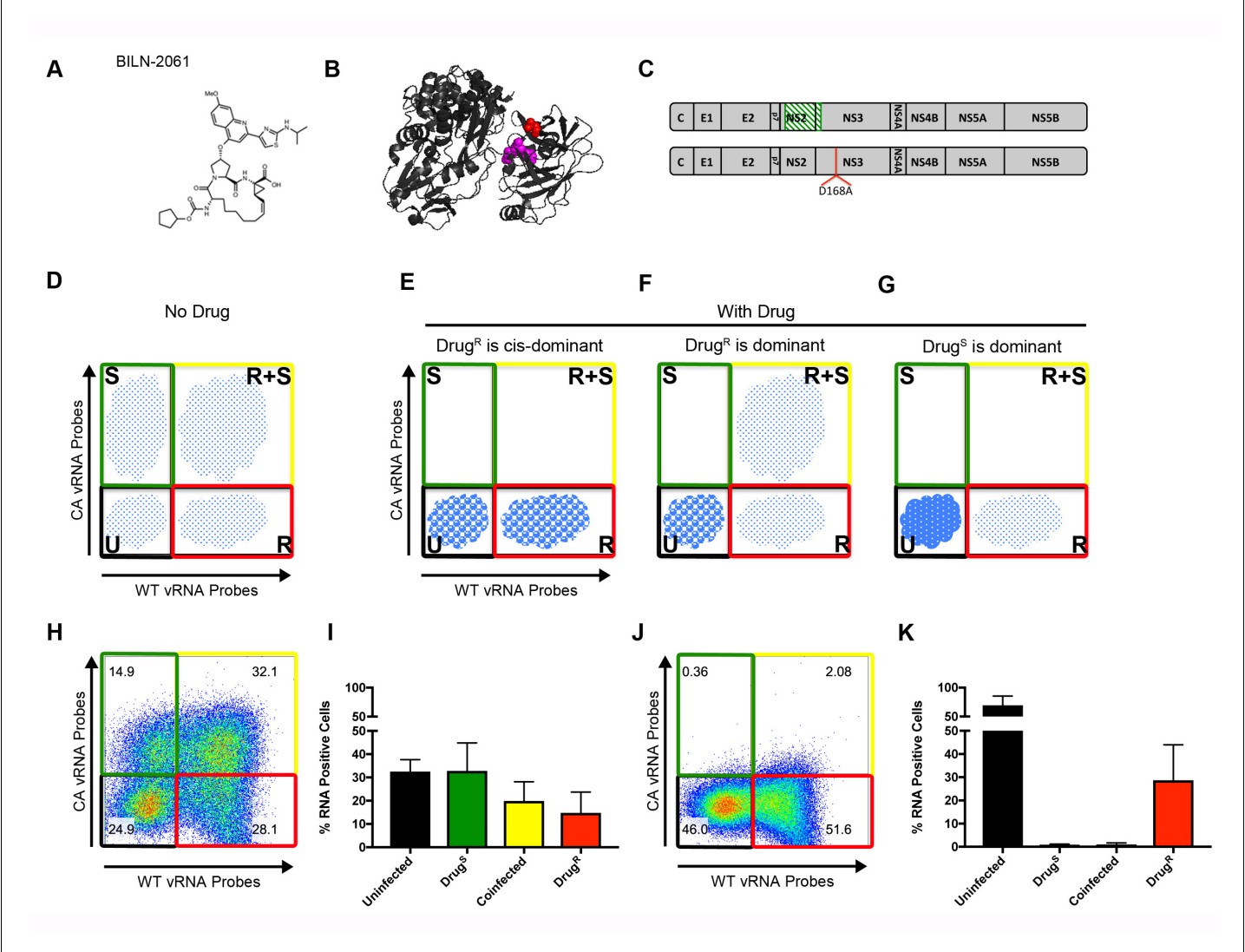

**Figure 3.** Flow cytometry to test dominance of viruses resistant to NS3 protease inhibitor. (**A**) Structure of protease inhibitor BILN-2061. (**B**) Structure of NS3 protein. D168 (red) is located in the protease domain adjacent to the active site (lavender). D168A confers resistance to BILN-2061. (**C**) Diagram of CA virus with altered sequence (green hatches) and WT virus with location of D168A mutation identified. (**D**) The four types of cells present in the absence of inhibitors are uninfected (U), infected with drug-susceptible virus cells (S), infected with both drug-susceptible and drug-resistant virus (S + R) and drug-resistant virus (R). In the presence of a DAA, three outcomes are possible and are indicated by the changing density of the cell populations: (**E**) Drug-resistance is *cis*-dominant, (**F**) Drug resistance is dominant and (**G**) Drug susceptibility is dominant. Huh7.5.1 cells were coinfected with CA and WT-D168A for 72 hr followed by treatment with (**H/I**) DMSO or (**J/K**) 2 μM BILN-2061 for 36 hr. Cells were stained with CA and WT vRNA probes and analyzed by flow cytometry (**H, J**) and results from three replicates quantified (**I, K**). NS3 drug-resistance was found to be *cis*-dominant.
DOI: https://doi.org/10.7554/eLife.32579.008

The following figure supplements are available for figure 3:

**Figure supplement 1.** NS3-D168A confers resistance to BILN-2061 in both the wild-type and the codon-altered backgrounds.
DOI: https://doi.org/10.7554/eLife.32579.009

**Figure supplement 2.** Exogenously expressed drug-resistant NS3 and NS5A do not rescue drug-susceptible HCV in trans.
DOI: https://doi.org/10.7554/eLife.32579.010

**Figure supplement 3.** Drug resistance to R1479 confers major fitness cost to JFH1.
DOI: https://doi.org/10.7554/eLife.32579.011

presence of 25 μM R1479. Several variants in NS5B (A336P, D438G, S282T, F427L, T481A) arose during passage (*Figure 3—figure supplement 3*). Each mutation was introduced independently into the JFH1 genome and RNA transfections were performed. The T481A genome was the only variant to show any viral RNA production by 7 or 21 days post-transfection. We noticed that F427L and T481A were always isolated together. To test whether these mutations could together increase viral fitness, JFH1 viruses were generated that contained both mutations. Viruses with the mutations separate or together were passaged extensively in the presence of R1479. Occasional resistant outgrowths were observed, but none conferred sustained growth (*Figure 3—figure supplement 3C*). Thus, like Sofosbuvir, the poor viability of mutant viruses resistant to R1479 precludes the ability to perform further genetic analysis but provides an excellent paradigm for antiviral development.

## Transmission genetics and phenotypic dominance of drug-resistant NS5A variant Y93N

NS5A is highly oligomeric (*Sun et al., 2015*; *Tellinghuisen et al., 2005*) and we were curious as to whether drug resistance or drug susceptibility would be dominant during viral infections. This idea seemed promising because exogenously expressed NS5A has a dominant-negative effect on the growth of HCV replicons (*Graziani and Paonessa, 2004*). Additionally, the NS5A inhibitors, as a class, display EC$_{50}$'s in the low picomolar range (*Gao, 2013*), making them among the most potent antiviral compounds ever identified. Assuming uniform inhibitor concentrations in cells and in medium, it has been estimated that only a small fraction of NS5A molecules should be bound to drugs under inhibitory conditions (*Sun et al., 2015*; *Gao et al., 2016*). Thus, it seemed mechanistically likely that drug-bound NS5A proteins from drug-susceptible viruses could be dominant inhibitors of NS5A encoded by newly arising drug-resistant ones. However, NS5A inhibitors have generally demonstrated low barriers to resistance in patients. Our goal was gain mechanistic insight into this dichotomy.

The structures of two such potent NS5A inhibitors, SR2486 (also known as BMS-346)(*Lemm et al., 2011*) and Daclatasvir (*Gao et al., 2010*) are shown in *Figure 4A*. Mutations of Tyr93 to Asp or His confer resistance to a broad array of NS5A inhibitors (*Gao et al., 2016*). Tyr93 is located near an NS5A dimer interface shown in the crystal structure (*Figure 4B*)(*Tellinghuisen et al., 2005*). Thus, this interface is postulated to be part of the binding site for the NS5A inhibitor class. The Y93N and Y93H mutations were introduced into both the wild-type and codon-altered viruses. As shown in *Figure 4C*, the Y93H mutation conferred resistance to both SR2486 and Daclatasvir while the Y93N mutation conferred resistance only to SR2486.

To test whether susceptibility to NS5A inhibitors was dominant in the context of viral infections, we analyzed U, S, S + R and R cell populations by flow cytometry as previously performed for the NS3/4A inhibitor in *Figure 3*. Huh-7.5.1 cells were coinfected with CA and WT-Y93N viruses for 72 hr (*Figure 4D*). Cells were then treated with DMSO or 500nM SR2486 for 24 hr, harvested, fixed, co-stained for WT and CA vRNAs and analyzed by flow cytometry. In the absence of the NS5A inhibitor, all four populations, U, S, R + S and R were observed (*Figure 4E,F*). In the presence of SR2486, the S population of cells dropped into the U population as expected. As was the case in *Figure 3*, the co-infected R + S population of cells dropped into the R population. Thus, resistance to NS5A inhibitor SR2486 in the context of viral infection was genetically dominant and the lack of rescue of the S virus with which it was mixedly infected shows that drug resistance is also *cis*-dominant. HCV infected cells become resistant to superinfection upon expression of non-structural proteins (*Schaller et al., 2007*; *Tscherne et al., 2007*). Due to this superinfection exclusion, it is likely that all coinfected cells arise through nearly synchronous infection throughout the course of the experiment. To control for any effects on selection that may occur due to the differential timing of coinfections that occurs over the initial 72 hr incubation period, we performed the same experiment with higher titer virus and a single cycle of infection in the absence of drug. Huh7.5.1 cells were infected at an MOI of 1 focus-forming unit (FFU)/cell with CA and WT-Y93N viruses and incubated for only 24 hr before drug treatment. Cells were then incubated in the absence and presence of 500nM SR2486 for an additional 24 hr. In this case, we also observe *cis*-dominance of drug resistant WT-Y93N genomes, indicating that asynchronous coinfection has no effect on selection (*Figure 4G*). Finally, the *cis*-dominance of Daclatasvir-resistant WT-Y93H was observed when coinfected with drug susceptible virus (S) in the absence and presence of Daclatasvir (*Figure 4H*). We conclude that NS5A, despite being an oligomeric species is not a dominant drug target. Instead, genomes resistant to

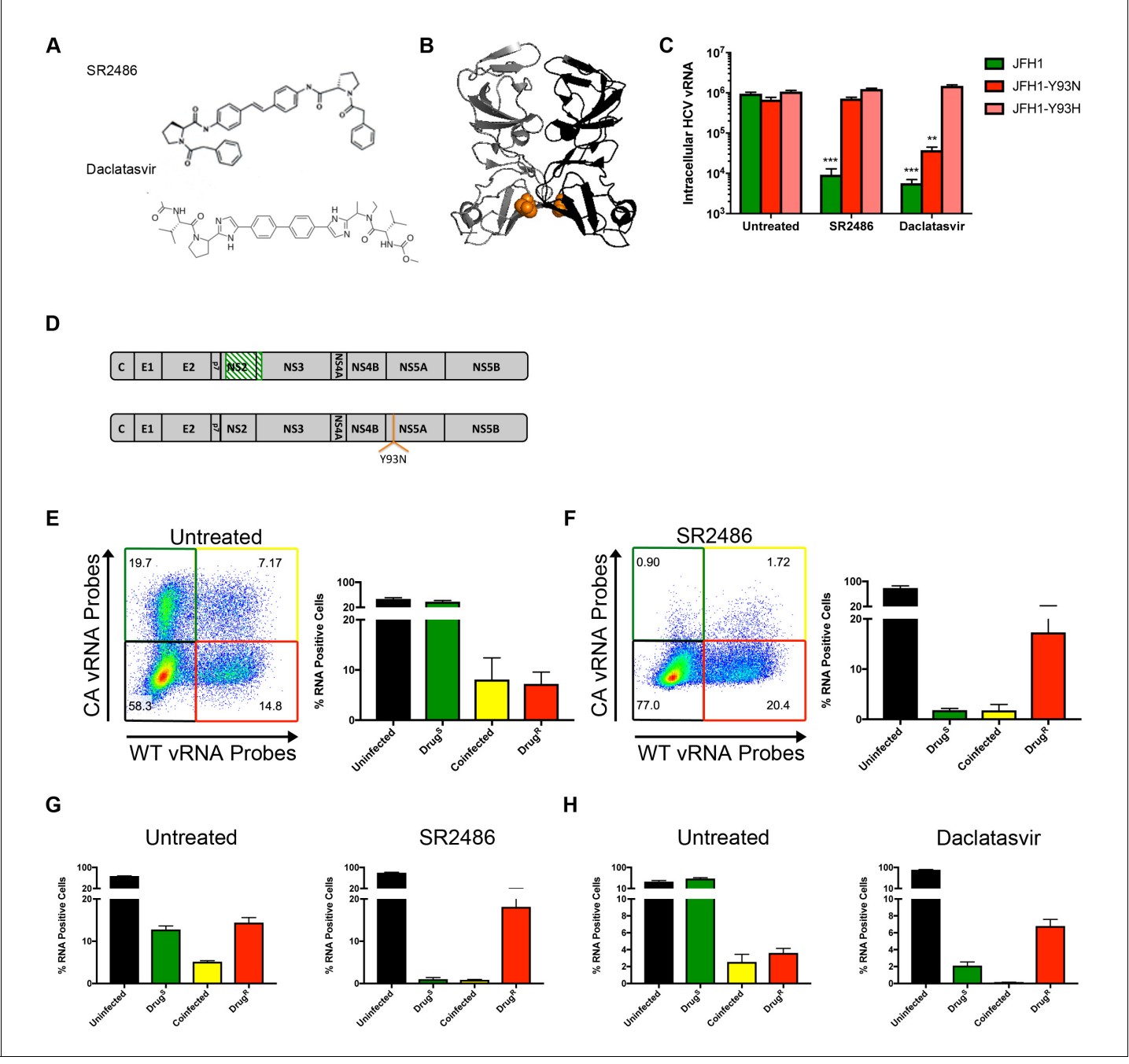

**Figure 4.** NS5A-resistant HCV is *cis*-dominant. (**A**) Structures of SR2486 or Daclatasvir. (**B**) Structure of NS5A dimer with Y93 identified (orange). NS5A variants, Y93N and Y93H have previously been shown to confer drug resistance to multiple NS5A inhibitors. (**C**) Huh7.5.1 cells were infected with WT, WT-Y93N or WT-Y93H at MOI of 0.1 FFU/cell in the absence or presence of 500nM SR2486 or 50 nM Daclatasvir to determine the drug resistance profiles. (**D**) Diagram of CA virus with altered sequence (green hatches) and WT virus with location of Y93N mutation identified. (**E/F**) Huh7.5.1 cells were coinfected with CA and WT-Y93N for 72 hr followed by treatment with (**F**) or without (**E**) 500nM SR2486 as indicated for 24 hr and analyzed by flow cytometry. Results from four replicates of the experiment shown are quantified. (**G**) Huh7.5.1 cells were infected with CA and WT-Y93N at a MOI of 1 for 24 hr followed by treatment with DMSO or 500nM SR2486. Resistance to SR2486 was found to be *cis*-dominant. (**H**) Results from three replicate experiments in which Huh7.5.1 cells were coinfected with CA and WT-Y93H for 72 hr followed by treatment without (left) or with 50 nM Daclatasvir (right) are shown. Resistance to Daclatasvir was found to be *cis*-dominant.

DOI: https://doi.org/10.7554/eLife.32579.012

NS5A remain drug resistant in co-infected cells but do not rescue drug-susceptible viruses present in the same cell. This is consistent with the observed outgrowth of viruses that are resistant to NS5A both in cultured cells and in patients, and with an earlier report that at least some functions of NS5A act exclusively in *cis* (*Fridell et al., 2011*).

One hypothesis that could mechanistically account for *cis*-dominant drug resistance is that NS5A molecules expressed from different alleles may not freely associate in mixed oligomers. As previously demonstrated, two different NS5As expressed from the same RNA can associate, while NS5A molecules expressed from different constructs could not (*Berger et al., 2014*). We were curious whether the dominance phenotypes were altered if we forced NS5A alleles to mix. To test whether exogenously expressed drug-susceptible NS5A proteins could co-assemble with drug-resistant NS5A, we utilized the previously described HCV plasmid that expresses HA-tagged and GFP-tagged NS5A within the same polyprotein but does not support genome replication (*Figure 5A*). Constructs that contained all combinations of drug-susceptible NS5A (S) and the drug-resistant Y93N variant (R) were created. Upon transfection, all tagged proteins were expressed and can be observed in *Figure 5* (Input, Panels B,C). Immunoprecipitation with anti-HA antibodies revealed that the GFP-tagged and HA-tagged NS5A proteins were present in the same complexes in the presence or absence of SR2486. Therefore, as has been shown previously, mixed oligomers can form upon co-expression within the same polyprotein (*Berger et al., 2014*). Furthermore, these interactions are not disrupted by drug treatment or by drug-resistant mutations (*Figure 5B,C*).

To determine whether there were any functional consequences of mixed oligomer formation, we visualized cells that expressed mixed oligomers using confocal microscopy. All S and R combinations of NS5A co-localized at discrete membrane-associated complexes characteristic of HCV infection in the absence of drug (*Figure 5D*, top panel). However, in the presence of SR2486, membrane-associated complex formation was inhibited in R:S and S:S expressing cells and observed only in R:R expressing cells (*Figure 5D*, bottom panel). The dispersal of NS5A signal upon drug treatment in the presence of S protein makes NS5A protein appear less abundant (*Figure 5D*). However, the immunoblots demonstrate that no such decrease in expression occurs as we observe equal levels of NS5A protein independent of allele or the presence of drug (*Figure 5B,C*). One hallmark of HCV infection is the accumulation of cytoplasmic lipid droplets (*Miyanari et al., 2007*; *Romero-Brey et al., 2012*). Electron microscopy performed by high-pressure freezing and freeze-substitution, to preserve membrane structure, revealed many lipid droplets in the cytoplasm of cells expressing S:S, R:R and S:R combinations of NS5As in the absence of inhibitor (*Figure 5E,F*). However, in the presence of SR2486, only the R:R cells displayed the accumulation of lipid droplets (*Figure 5E, G*). Therefore, using both assays, the presence of drug-susceptible NS5A prevented drug-resistant phenotypes from being displayed, and thus drug-susceptibility was genetically dominant. This confirmed our original hypothesis that NS5A had the potential to be a dominant drug target.

## Lack of free mixing may prevent NS5A hetero-oligomerization

One of the most likely mechanisms for *cis*-dominance, when the benefit of a particular gene product accrues only to the genome that encodes it, is physical isolation. We hypothesized that HCV genomes co-infecting the same cell might be physically isolated from each other. To test this possibility, confocal microscopy was used to identify and localize negative vRNA strands in genome-specific RNA replication complexes (*Figure 6A*). The majority of negative strands of the two different viruses were found to be discrete. Identification and quantification the vRNA puncta in coinfected cells was determined computationally using Volocity software. This program determined the number of negative strand puncta per cell per strain and quantified how many puncta overlapped (*Figure 6B*). This value was low even for the positive stranded vRNAs, which are present in the cytoplasm at much higher frequencies (*Figure 6A,B*).

As a positive control for colocalization, we performed a similar experiment but additionally stained for NS5A or Core in addition to minus strand vRNAs. We would expect minus strand vRNA and NS5A to colocalize strongly, as NS5A is present inside replication complexes. Alternatively, Core is not localized directly within replication complexes, but is present within packaging complexes and on lipid droplets, which are nearby. Volocity was used to count negative strand vRNAs, and then asked, how many of those puncta were touching NS5A or Core. Representative images demonstrating each of the pairwise comparisons demonstrate that nearly 80% of all minus strand

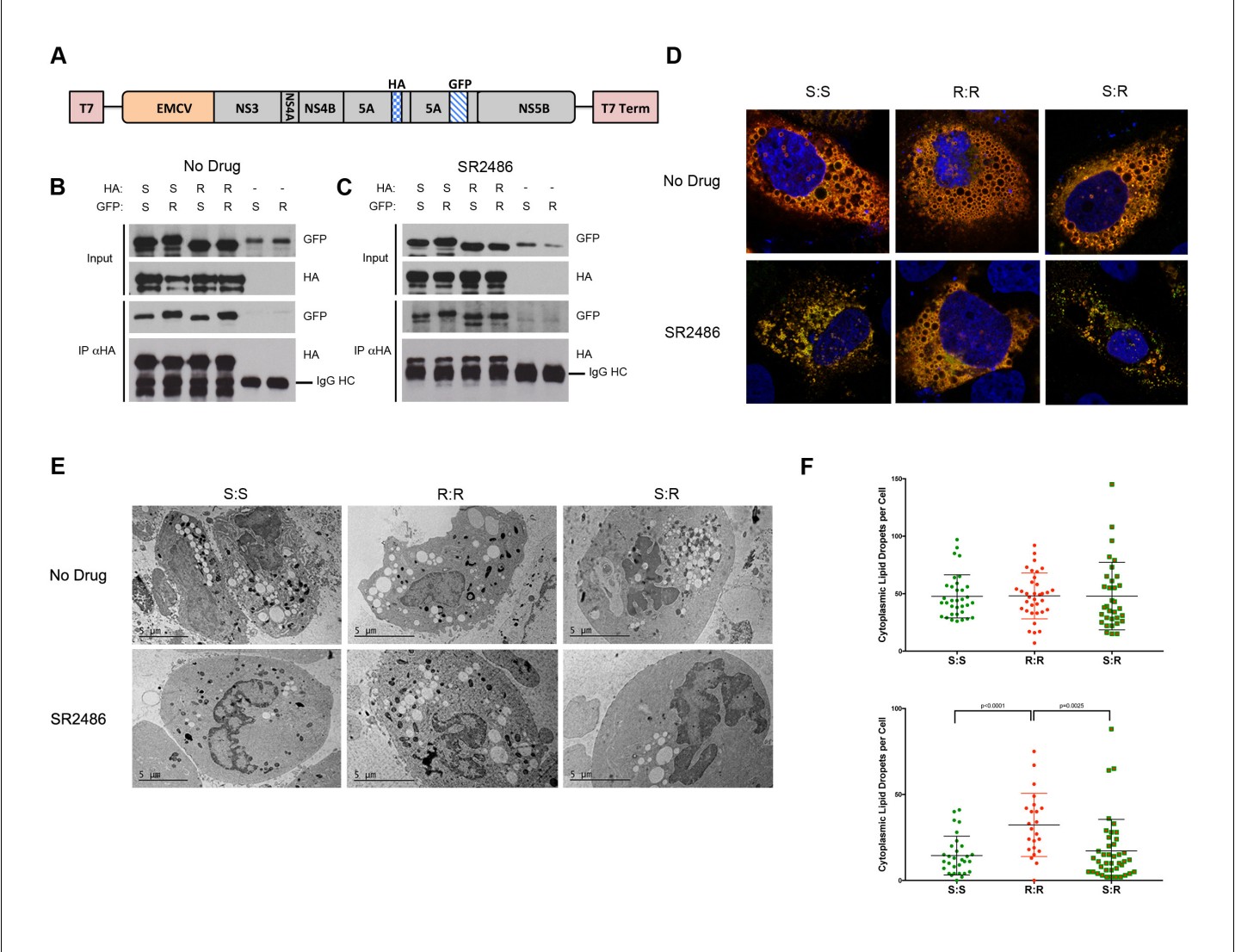

**Figure 5.** Complex formation between drug susceptible and drug-resistant NS5A in the absence of RNA replication. (A) HCV NS3-5B constructs driven by T7 polymerase and encoding tandem tagged copies of NS5A (*Berger et al., 2014*) were created to co-express HA- and GFP-tagged NS5A alleles. Huh7-Lunet-T7 cells were transfected with pTM-Dual-NS5A constructs that contained two drug-resistant (R), two drug susceptible (S), or mixed alleles of NS5A. Proteins from transfected cell extracts that were incubated in the absence (B) or presence (C) of SR2486 were subjected to SDS-PAGE without further fractionation (Input) or after immunoprecipitation with anti-HA antibodies (IP αHA). The gel was subjected to immunoblotting with GFP or HA antibodies as indicated. (D) Cells were transfected with dual-NS5A constructs in the absence or presence of SR2486 for 24 hr. Cells were then fixed, stained with anti-HA antibodies and visualized by confocal microscopy. Representative images from over 25 cells are presented. (E) Cells transfected with dual-NS5A constructs that expressed drug-susceptible (S) and drug-resistant (R) alleles as shown were prepared for electron microscopy by high-pressure freezing and freeze-substitution and visualized by transmission electron microscopy. (F) Numbers of cytoplasmic lipid droplets per cell formed in the absence (left) or presence (right) of SR2486; at least 25 images per sample such as those shown in (E) were quantified.
DOI: https://doi.org/10.7554/eLife.32579.013

vRNAs were touching NS5A while fewer than half of the minus strand vRNAs were touching Core (*Figure 6C,D*).

These data support the hypothesis that, upon co-infection, drug-resistant and drug-susceptible RNA genomes create independent membranous web structures, limiting the mixing of NS3/4A and NS5A proteins and their precursors. This scenario is modeled schematically in *Figure 6E*. Failure of NS5A proteins to mix during infection is a likely explanation for the *cis*-dominance of drug resistance observed in cultured cells (*Figure 4*). These circumstances account for the ready outgrowth of drug resistance in patients (*Gao et al., 2016*), even though NS5A is highly oligomeric.

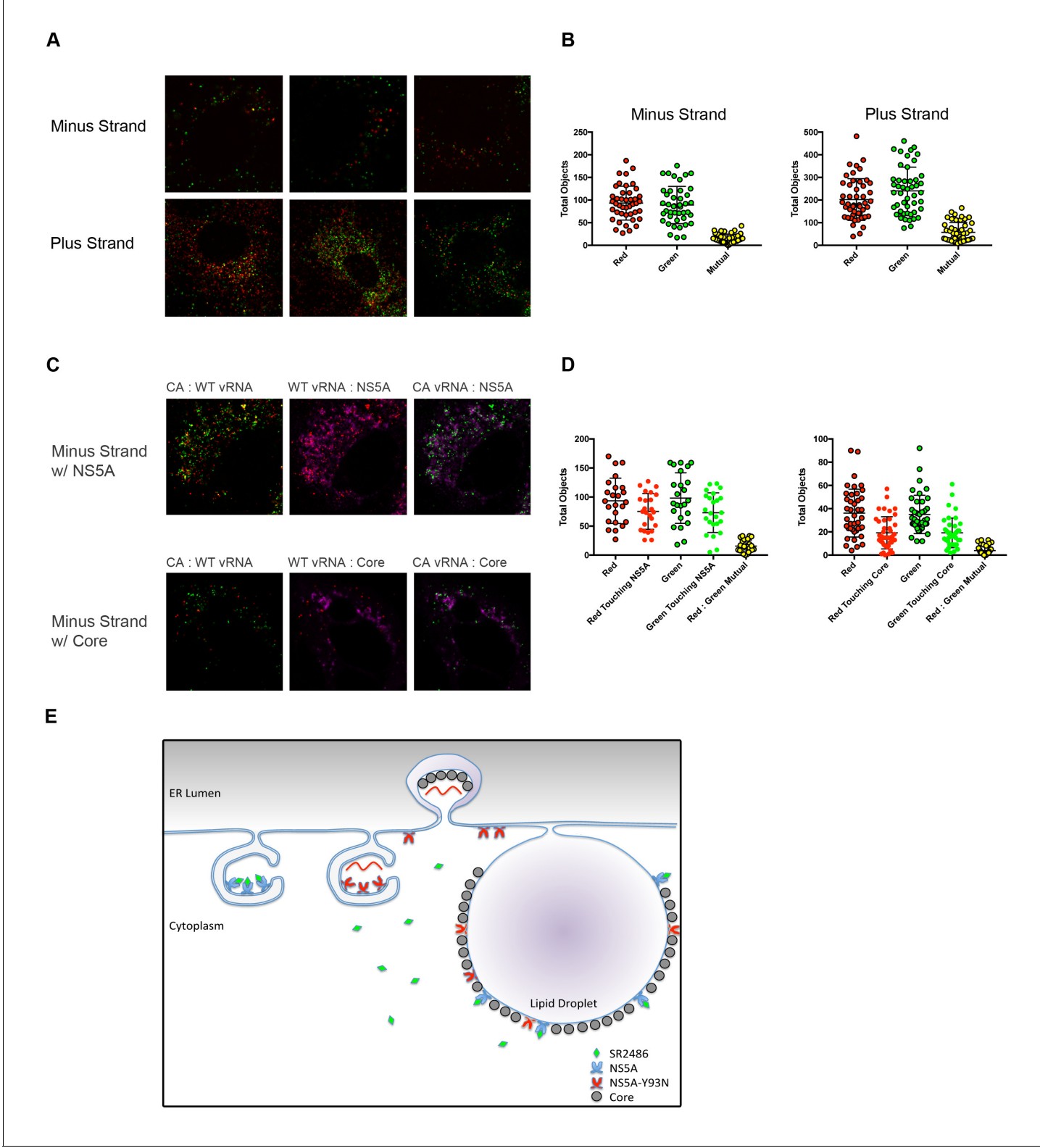

**Figure 6.** Drug-susceptible and drug-resistant RNA replication complexes segregate in coinfected cells. (**A**) Huh7.5.1 cells were coinfected with CA and WT-Y93N at a MOI of 1 FFU/cell for 24 hr. Cells were fixed and co-stained with WT and CA negative-strand or positive strand viral RNA probe sets and visualized by confocal microscopy. Co-infected cells were identified and three representative images are displayed of more than 50 captured images. (**B**) Velocity was used to identify and quantify vRNA puncta within coinfected cells. These puncta were then assessed for colocalization and quantified.
*Figure 6 continued on next page*

*Figure 6 continued*

(C) Huh7.5.1 cells were coinfected with CA and WT-Y93N at a MOI of 1 FFU/cell for 24 hr. Cells were costained to visualize Core or NS5A together with CA and WT vRNAs. Representative cells are displayed demonstrating all pairwise comparisons analyzed for colocalization. (D) Volocity was used to quantify the number of vRNAs per cell, the number of colocalized vRNAs, as well as the number of vRNA puncta touching NS5A or Core. (E) Depiction of the clonal nature of individual RNA replication sites (Adapted from Figure 9 in *Zayas et al., 2016*]). Membrane invaginations house either drug-resistant (red) or drug-susceptible (blue) genomes. In the model, the RNA replication sites are segregated and therefore only the RNA from drug-resistant virus is amplified in the presence of inhibitors of RNA replication. It is visually suggested that NS5A molecules that bring core protein to lipid droplets for viral assembly mix on this surface and that this could lead to genetic dominance of drug susceptibility at a packaging step.

DOI: https://doi.org/10.7554/eLife.32579.014

Even when drug-resistant NS5A was overexpressed in a precursor form, no rescue of drug-susceptible virus was detected (*Figure 3—figure supplement 2C*). It has previously been shown that when HA- and GFP-tagged NS5A molecules were expressed on different constructs such as those depicted in *Figure 5A*, no mixed complexes were formed (*Berger et al., 2014*). Thus, even though high-order NS5A oligomers are formed in infected cells, it is unlikely that these are mixed-allele oligomers, preventing dominant inhibition of drug-resistant HCV.

## Discussion

Due to the highly mutagenic nature of RNA viruses and the large number of genomes and antigenomes generated during infection, a high barrier to drug resistance is extremely difficult to achieve. This has led to abandoned usage and development of many otherwise promising antivirals. To decrease the frequency with which drug-resistant variants arise, combinations of antivirals that, individually, exhibit low barriers to resistance are often used. When drug-resistant variants are first formed intracellularly, through error-prone RNA replication, they arise in a population that includes parental and sibling drug-susceptible viruses. Several genetic relationships between drug-resistant and drug-susceptible genomes are possible. First, the drug resistance of the new variants has the potential to be genetically dominant, and rescue both resistant and susceptible viral genomes. Alternatively, drug resistance can be *cis*-dominant, with the drug-resistant products rescuing only the genomes that encode them. Finally, the drug-resistant genome can fail to benefit any genomes in the cell because the drug-susceptible products present in the same cell are dominant inhibitors.

The DAAs targeting NS3/4A protease of HCV were the first to be discovered (*Lamarre et al., 2003*) and the first to reach the clinic (*Bacon et al., 2011*; *Jacobson et al., 2011*). It was soon realized that, both during the growth of HCV replicons in cultured cells and in phase II clinical trials, drug-resistant viruses were generated rapidly (*Lin et al., 2004*). Nonetheless, in 2011, further advances led to FDA approval of Telaprevir and Boceprevir (*Jacobson et al., 2011*; *Poordad et al., 2011*). The anticipation, which proved to be correct, was that inhibitors of NS3/4A would prove useful in combination therapies (*Feld et al., 2014*; *Lawitz et al., 2014*). We have used flow cytometry to identify cell populations that are co-infected with HCV that is susceptible or resistant at a 1:1 ratio, to protease inhibitors. Within these cells, the drug-resistant genomes replicated, but the drug-susceptible genomes did not. We therefore conclude that NS3/4A inhibitor resistance is *cis*-dominant (*Figure 3*), which should allow the rapid and specific selection for outgrowth from its cell of origin.

*Cis*-dominance of drug resistance was also not originally anticipated while targeting NS3/4A. The original characterizations of the NS3/4A protease suggested that cleavage of the NS3/4A junction occurred in *cis*, but that cleavages at the 4A/4B, 4B/5A and 5A/5B junctions could all occur in *trans* (*Bartenschlager et al., 1994*). We felt that it was therefore, more likely, that drug-resistant NS3/4A could rescue drug-susceptible virus within the same cell. NS3/4A is not known to assemble into high-order oligomers in the same manner as NS5A, and we therefore did not anticipate drug-susceptible NS3/4A would be *trans*-dominant. Furthermore, a *trans* cleavage assay demonstrated that a NS4B-5B polyprotein could be cleaved by NS3/4A supplied in *trans* (*Romero-Brey et al., 2015*). However, the *trans*-cleavage system does not result in membranous web formation that would accompany genome sequestration. Other groups have reported a different result, that defective NS3 mutants cannot be rescued in *trans* by replicons with functional NS3/4A (*Kazakov et al., 2015*; *Appel et al., 2005*). Our interpretation of these studies is that NS3/4A is likely physically able to

cleave in *trans* in cells, but requires access to the alternate precursor proteins in order for this to occur. Therefore, *cis*-dominance of drug-resistance is likely the result of a lack of free-mixing of NS3/4A encoded by different vRNAs within the same cell.

NS5A emerged as an HCV drug target through a chemical genetics screen for compounds that inhibited HCV growth but did not target the NS3/4A protease or the NS5B polymerase (*Gao et al., 2010*). The ease with which resistant viruses were selected suggested that drug resistance was either dominant or *cis*-dominant. This was somewhat surprising, given that NS5A is oligomeric and the NS5A inhibitors are extremely potent, and have been postulated to function at sub-stoichiometric ratios to NS5A protein (*Gao, 2013*). Indeed, when drug-susceptible and drug-resistant NS5A protein were co-expressed in the present study, hetero-oligomers formed and the biological phenotypes of the drug-susceptible protein were dominant (*Figure 5*). Nonetheless, single-cell analysis of cells co-infected at a 1:1 ratio with NS5A inhibitor-susceptible and -resistant viruses showed, as with the NS3/4A inhibitor, that resistance to both SR2486 and Daclatasvir was *cis*-dominant (*Figure 4*).

What does *cis*-dominant resistance mean mechanistically? One potential mechanism is physical sequestration of the RNA replication complexes of the two co-infecting genomes. Genome-specific RNA probing of co-infected cells revealed that both the negative strands and positive strands from the two viruses were present at physically distinct locations (*Figure 6*). It is therefore highly probable that membrane-associated proteins such as HCV NS3/4A and NS5A do not mix within individual RNA replication complexes. However, not all mutations in a particular viral product should lead to the same defect with the same genetic properties. For example, we show here that viruses that are defective in a function of NS5A in RNA replication complexes are not rescued and have no effect on the outgrowth of drug-resistant variants. However, NS5A also plays an important role in packaging and assembly of mature HCV particles on lipid droplets (*Miyanari et al., 2007*; *Boson et al., 2017*). Lipid droplets are large and form adjacent to RNA replication complexes. In *Figure 6E*, we have depicted the possibility that NS5A molecules encoded by distinct RNA replication complexes might mix on the surface of these lipid droplets. However, if replication of the drug-susceptible genomes is inhibited, contribution of their encoded proteins to any oligomers on the surface of lipid droplets should be minimal. In this vain, a hypothetical NS5A inhibitor that allowed RNA replication but inhibited the function of NS5A in particle assembly might have different genetic properties than the NS5A inhibitors currently in use.

Viral capsids have especially interesting genetic properties, often intermixing within co-infected cells. Defective capsid proteins of poliovirus, HBV and HIV have been shown to be dominant inhibitors of wild-type viruses (*Crowder and Kirkegaard, 2005*; *Tanner et al., 2016*; *Tan et al., 2013*; *Tan et al., 2015*; *Trono et al., 1989*; *Pettit et al., 2005*; *Lee et al., 2009*; *Müller et al., 2009*; *Checkley et al., 2010*). Thus, when antiviral targets are capsid proteins, drug susceptibility can be genetically dominant by suppressing the outgrowth of drug-resistant virus within the cell in which it is first generated (*Crowder and Kirkegaard, 2005*; *Tanner et al., 2014*; *Kirkegaard et al., 2016*). For HCV, very few inhibitors of capsid function have been identified, and their inhibition of viral growth is not sufficiently robust to make genetic experiments possible (*Kota et al., 2010*). It is therefore not yet possible to test if, as we hypothesize, drug-susceptible virus will prove to be a dominant inhibitor of drug resistance. Consistent with this hypothesis, however, epitope-tagged HCV core protein can form mixed disulfide-bonded core oligomers (*Kushima et al., 2010*).

The success of combination therapy for HCV and the efficacy of the individual constituents illustrate some of the weapons in the arsenal of antiviral strategies. Future directions are likely to include, as well, the rational design of antivirals with high barriers to resistance such as those that hyper-stabilize oligomers and the prediction of DAA targets that impart a high fitness cost to drug resistance.

## Materials and methods

### Cells and viruses

Huh7.5.1 cells were a gift from Dr. Michael Gale Jr (University of Washington) and were cultured in DMEM (Sigma, St. Louis, MO) supplemented with 10% fetal bovine serum (Omega, Tarzana, CA), penicillin/streptomycin (Invitrogen, Grand Island NY), non-essential amino acids (Invitrogen), and Glutamax (Invitrogen). Huh7-Lunet-T7 cells were a gift from Dr. Ralf Bartenschlager (University of

Heidelberg) and were cultured in DMEM supplemented with 10% fetal bovine serum, penicillin/streptomycin, non-essential amino acids, Glutamax and 5 µg/mL Zeocin (Invitrogen). Cell line identification was performed using STR profiling services available through the Stanford Functional Genomics Facility. Alignments were generated using Huh7 as a reference. Cell lines were screened for mycoplasma contamination using the MycoAlert Mycoplasma Detection Kit (Lonza).

The plasmid pJFH1 was a gift from Dr. Michael Gale Jr (*Kato et al., 2006*). This plasmid contains a synthesized genome length copy of the JFH1 strain of HCV (genotype 2a). To produce cell-culture-derived HCV particles (HCVcc), pJFH1 was digested with *XbaI* (New England Biolabs). The linearized plasmid was then used as a template for in vitro transcription with the MEGAscript high yield transcription kit (Ambion). vRNA was purified using Trizol (Invitrogen) and electroporated into Huh7.5.1 cells as previously described to generate HCVcc cultures (*Wakita et al., 2005*). Following a period of amplification, HCVcc cultures were converted to human serum media as described previously (*Steenbergen et al., 2013*). Human serum media comprised DMEM supplemented with 2% heat inactivated human serum (Omega), penicillin/streptomycin, non-essential amino acids and Glutamax.

## Antibodies

Antibodies recognizing HCV core (Abcam), GAPDH (Santa Cruz Biotechnologies), GFP (Life Technologies) and HA (Genscript) were purchased from the individual suppliers. Antibodies recognizing NS5A were described previously (*Lindenbach et al., 2005*).

## HCVcc constructs

To construct codon-altered strains of HCV, we subjected three approximately 1000-nucleotide fragments of the JFH1 genome through the GeneArt codon optimization algorithm offered by Life Technologies. The genome fragments were composed of nucleotides 2613–3530 (CA-3), 7441–8456 (CA-2), and 7867–8896 (CA-1). All three codon-altered genome fragments were synthesized by Life Technologies and cloned into the pJFH1 plasmid by restriction digestion and ligation with T4 DNA ligase (Invitrogen). The resulting plasmids: pJFH1-CA-1, pJFH1-CA-2 and pJFH1-CA-3, were used to produce HCVcc cultures as described above.

To create drug-resistant HCVcc cultures, two subcloning plasmids were created by PCR by amplifying nucleotides 6395–8670 or 4584–6498 of the pJFH1 plasmid with Taq polymerase (New England Biolabs) and ligating the PCR products into pCR2.1 (Invitrogen). The resulting plasmids, pCR2.1-6395-8670 and pCR2.1-4584-6498 were used as templates for site-directed mutagenesis using the QuikChange Site-Directed Mutagenesis kit (Agilent Technologies). pCR2.1-6395-8670-Y93N was generated using the forward primer 5′-CCTATCAATTGC**AAT**ACGGAGGGCCAGTGCGCGCC-3′ and the reverse primer 5′-GGCGCGCACTGGCCCTCCGT**ATT**GCAATTGATAGG-3′. pCR2.1-6395-8670-Y93H was generated using the forward primer 5′-CCTATCAATTGC**CAT**ACGGAGGGCCAGTGCGCGCC-3′ and the reverse primer 5′-GGCGCGCACTGGCCCTCCGT**ATG**GCAATTGATAGG-3′. pCR2.1-4584-6498-D168A was generated using the forward primer 5′-AAATCCATC**GCC**TTCATCCCC-3′ and the reverse primer 5′-GGGGATGAA**GGC**GATGGATTTGGC-3′. These mutated HCV genome fragments were cloned into pJFH1 or pJFH1-CA using restriction digestion and ligation with T4 DNA ligase (Invitrogen). HCVcc cultures were generated as described above.

## Plasmids

The plasmids pTM_NS3-5B_NS5A-HA_2a_NS5A-gfp_JFH1 (referred to as pTM-Dual-NS5A) and pTM_NS3-5B_NS5A-GFP (referred to as pTM-NS3-5B) were the generous gifts of Dr. Ralf Bartenschlager (University of Heidelberg). The D168A and Y93N mutations were cloned into the pTM-NS3-5B plasmid using the Quikchange Lightening Mutagenesis kit using the primers described above. The NS5A alleles of the pTM-Dual-5A plasmid were first separated by removing an RsrII fragment containing most of the NS5A-GFP allele to create pTM-Dual-5A-ΔRsrII and pcDNA5-NS5A-GFP-RsrII. Site-directed mutagenesis was performed using the Quikchange Lightening kit on pTM-Dual-5A-ΔRsrII or on pcDNA5-NS5A-GFP-RsrII independently. The RsrII fragments containing wild-type NS5A or NS5A-Y93N were then cloned back into the pTM-Dual-5A-ΔRsrII vectors to create all combinations of wild-type NS5A and NS5A-Y93N pTM-Dual-5A.

## qRT-PCR

vRNA was harvested from cells using Trizol (Invitrogen) or collected from HCVcc culture supernatants using the QIAamp vRNA mini kit (Qiagen). A standard curve was generated using *in vitro* transcribed HCV vRNA. qRT-PCR was performed using the QuantiTect Sybr-Green RT-PCR kit (Qiagen) and the qRT-PCR forward 5'-CTGGCGACTGGATGCGTTTC-3' and reverse 5'-CGCATTCCTCCATC TCATCA-3' primers. Alternatively, the following CA-specific primers were used: forward 5'-GTGGTG TCCATGACCGGCA-3' and reverse 5'-GGTCACGGGGCCTCTCAGT-3', or the following WT-specific primers were used: forward 5'-GTGGTGAGTATGACGGGGC-3' and reverse 5'-CGTGACCG-GACCCCGTAAG-3'. Samples were analyzed on a 7300 Real-Time PCR Machine (Applied Biosystems).

## Confocal microscopy

WT vRNA target probes recognizing the NS2 region of either the positive or negative strand were designed and synthesized by Affymetrix. These probes were specifically designed to avoid detection of codon-altered JFH1 viral RNA. Additionally, probes were designed to recognize the corresponding region of the negative or positive strand JFH1-CA vRNA. These CA target probes were specifically designed not to recognize the WT vRNA.

Huh7.5.1 cells were infected with WT or CA HCVcc particles for 72 hr. Infected cells were fixed with 4% formaldehyde solution (Sigma) and subjected to RNA *in situ* hybridization (ISH) using the ViewRNA Cell Assay kit (Affymetrix) according to the manufacturer's protocol. Cells were co-stained with both CA and WT vRNA target probe sets in all experiments. Cells were visualized on a Leica SP8 confocal microscope. Protein and vRNA colocalization was performed on cells coinfected with JFH1-CA and JFH1-Y93N for 24 hr. Following infection, cells were fixed and stained using the View-RNA Cell Plus assay reagents. Core and NS5A were visualized using the antibodies described above at a 1 to 100 dilution followed by the anti-mouse-AlexaFlour-647 secondary antibody at 1 to 200 dilution.

Quantification of colocalization was performed using Volocity software (Perkin Elmer). Briefly, we defined vRNA puncta as objects larger than 0.1 $\mu m^2$. Objects larger than 0.25 $\mu m^2$ were broken into subunits based on total volume. Objects sharing 0.05 $\mu m^2$ of mutual space were quantified as mutual. Due to the localization patterns of core and NS5A, spot counting algorithms were not appropriate. Total vRNA objects and as well as the total number of vRNA objects touching NS5A or Core were quantified.

Huh7-Lunet-T7 cells were transfected with pTM-Dual-NS5A constructs using branched polyethylenimine (Sigma-Aldrich) at a ratio of 1:3. At 4 hr post-transfection, cells were treated with 500nM SR2486 or a DMSO control. At 24 hr post-transfection, cells were fixed with 4% paraformaldehyde, stained with anti-HA antibodies and DAPI and visualized on a Leica SP8 confocal microscope. A more detailed description of the RNA FISH microscopy methods are provided in Bio-Protocol (*van Buuren and Kirkegaard, 2018*).

## Electron microscopy

Huh7-Lunet-T7 cells were transfected with pTM-Dual-NS5A constructs using the polyethylenimine transfection reagent. At 4 hr post-transfection, cells were treated with DMSO or 500nM SR2486. At 24 hr post-transfection, cells were harvested using an enzyme-free cell dissociation buffer (Life Technologies) and FACS sorted for GFP-positivity on a FACS Aria cell sorter. GFP-positive cells were resuspended in 20% BSA in PBS then placed into a 200 μM deep hat and high-pressure frozen using a Leica EMpact2. Frozen samples were then freeze substituted in 1% Osmium tetroxide and 0.1% uranyl acetate in acetone using a Leica EMAFS at −90°C for 72 hr, warmed to −25°C in 16.3 hr at 4°C/hr and held for 12 hr then warmed to 0°C in 5 hr at 5°C/hr and held for 12 hr. The samples were then washed two times in acetone, then in propylene oxide for 15 min each. Samples are infiltrated with EMbed-812 resin (EMS Cat#14120) mixed 1:2, 1:1, and 2:1 with propylene oxide for 2 hr each, leaving samples in 2:1 resin to propylene oxide overnight rotating at room temperature. The samples are then placed into EMbed-812 for three hours then placed into TAAB capsules with fresh resin and placed into a 65°C oven overnight.

Sections were taken between 75 and 90 nm, picked up on formvar/carbon coated 100 mesh copper grids, then contrast stained for 30 s in 3.5% uranyl acetate in 50% acetone followed by staining

in 0.2% lead citrate for 3 min. Cells were visualized using the JEOL JEM-1400 120kV microscope and photos were taken using a Gatan Orius 4k × 4k digital camera.

## Flow cytometry

Huh7.5.1 cells were either transfected with WT and/or CA vRNA as previously described or infected with WT and/or CA HCVcc particles. Coinfections were performed by infecting with each virus for 24 or 72 hr followed by treatment with either 2 μM BILN-2061 for 36 hr or 500nM SR2486 for 24 hr. Cells were harvested with trypsin and fixed with the FlowRNA Fixation and Permeablization kit. Cells were then costained with CA and WT vRNA target probe sets using the FlowRNA kit (Affymetrix) and analyzed on the Scanford FACScan Flow Cytometer. Data was analyzed and processed using Flowjo software.

## Acknowledgements

We thank Drs. Yury Goltsev and Garry Nolan for advice on fluorescent cell sorting-based visualization of RNA, Drs. Michael Gale Jr. and Ralf Bartenschlager for the generous donation of reagents and Dr. Peter Sarnow for critical reading of the manuscript. We would like to acknowledge the work of Drs. Jeannie Spagnolo and Ernesto Mendez for initiating HCV research in our laboratory. This work was supported by funding to KK from NIH U19AI109662 (Jeffrey Glenn, PI), an NIH Director's Pioneer Award and the Alison and Steve Krausz Innovation Fund. NvB was supported by the Canadian Institutes for Health Research NCRTP-HepC training program and the American Liver Foundation. Electron microscopy was performed in a facility supported in part by ARRA award number 1S10RR026780-01 from the National Center for Research Resources (NCRR). The Cell Sciences Imaging Facility used for confocal microscopy was supported by ARRA award number 1S10OD010580 from the NCRR. The contents of this manuscript are solely the responsibility of the authors and do not necessarily represent the official views of the NCRR or the National Institutes of Health.

## Additional information

### Competing interests

Karla Kirkegaard: Reviewing editor, *eLife*. The other authors declare that no competing interests exist.

### Funding

| Funder | Grant reference number | Author |
| --- | --- | --- |
| National Institutes of Health | U19-AI09662 | Karla Kirkegaard |
| Canadian Institutes of Health Research | NCRTP-HepC Postdoctoral Fellowship | Nicholas van Buuren |
| American Liver Foundation | Postdoctoral Fellowship | Nicholas van Buuren |
| National Institutes of Health | NIH Director's Pioneer Award | Karla Kirkegaard |
| Alison and Steve Krausz Innovation Fund | | Karla Kirkegaard |

The funders had no role in study design, data collection and interpretation, or the decision to submit the work for publication.

### Author contributions

Nicholas van Buuren, Conceptualization, Data curation, Formal analysis, Funding acquisition, Validation, Investigation, Visualization, Methodology, Writing—original draft, Writing—review and editing; Timothy L Tellinghuisen, Resources, Provided compounds in advance of publication; Christopher D Richardson, Supervision, Funding acquisition, Writing—review and editing; Karla Kirkegaard,

Conceptualization, Resources, Supervision, Funding acquisition, Methodology, Writing—review and editing

### Author ORCIDs
Karla Kirkegaard http://orcid.org/0000-0001-7628-3770

### Decision letter and Author response
Decision letter https://doi.org/10.7554/eLife.32579.017
Author response https://doi.org/10.7554/eLife.32579.018

## Additional files

### Supplementary files
• Transparent reporting form
DOI: https://doi.org/10.7554/eLife.32579.015

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
