## [Decision Letter]

Thank you for submitting your article "Transmission genetics of drug-resistant hepatitis C virus" for consideration by *eLife*. Your article has been reviewed by three peer reviewers, and the evaluation has been overseen by a Reviewing Editor and Arup Chakraborty as the Senior Editor. The following individual involved in review of your submission has agreed to reveal his identity: 1); Matthew Evans (Reviewer #3).

The reviewers have discussed the reviews with one another and the Reviewing Editor has drafted this decision to help you prepare a revised submission.

Summary:

All three reviewers found this work to be very interesting and thought-provoking. The re-coding of a genome to allow FISH discrimination of the replication of two viral strains is very innovative. The idea of cis-dominant drug resistance is novel, and this may be explained by the segregation of different viral genomes into distinct replication complexes. There was a concern that the low MOI used may allow asynchronous infection of cells and lead to the segregation of the complexes at separate sites in the infected cell. The reviewers felt that the manuscript would also be improved by further discussion of the possible mechanisms of cis-complementation and the literature on the genetic interactions of interacting HCV genomes.

Essential revisions:

1) An experiment at higher MOI to confirm that the cis-dominance and segregation of the replication complexes are not due to asynchronous infection of cells. Even if it is not possible to use a high enough MOI to co-infect every cell, it should be possible to increase the level of co-infection. The authors could then make an argument as to why the MOI is not the cause of the segregation.

2) Further discussion of the literature on complementation between HCV viruses and further discussion of the mechanism of the cis-dominance would help the manuscript.

*Reviewer #1:*

1) The quality of the scholarship needs improvement. Specifically, the authors didn't cite some relevant publications that do not fit precisely with their model.

2) The use of low MOIs greatly complicates interpretation of the two color FACS data. First, very few cells will be doubly infected during early times of infection at low MOIs; most double positive cells will therefore be the product of asynchronous, secondary infections. Second, because HCV exhibits superinfection exclusion, these co-infections must occur very close in time. The result is that even small differences in fitness may allow one genotype to appear dominant over another simply because it spreads faster and occupies a subset of cells, rather than because of cis dominant drug sensitivity. Thus, replication phenotypes would have to be closely matched for this experimental design to work as reported. This seems most relevant to Figure 6, where the drug-resistant genotype replicated far less efficiently than the wild-type, drug-sensitive genotype (Figure 6C, left panel). If indeed S virus spread more efficiently, it would have prevented coinfection by R virus, sealing its fate. Conversely, if R had replicated more efficiently it would prevented coinfection with S virus, and would not have been burdened by supporting S.

3) Subsection “Transmission genetics and phenotypic dominance of drug-resistant NS5A variant Y93N”, fourth paragraph and Figure 5C. It appears that both S and R NS5A levels were greatly reduced when the S genotype was present during drug treatment. In other words, the expression of the S allele poisoned the expression of the R allele. I recommend that the authors point this out here. Also, this observation complicates the interpretation of Figure 7 that phenotypic mixing does not occur.

4) Subsection “Lack of free mixing may prevent NS5A hetero-oligomerization”, last paragraph and Figure 3—figure supplement 2. The reason why expression of the R allele was unable to rescue the S genome is trivial: DNA transfection renders Huh-7 cells resistant to HCV replication (i.e., the fraction of cells that are productively transfected with DNA are greatly resistant to HCV replication, likely due to induction of an innate antiviral response – go ahead and confirm this with your DNA of choice). Therefore, I expect that very little R was expressed in the same cells as the virus. The obvious approach would be to show a positive control complementation with the same technique used here.

5) Figure 7A. This panel is difficult to interpret for several reasons. First, related to point 2, above, we don't know whether these co-infections occurred simultaneously or sequentially. Second, while we don't know the efficiency of negative- strand detection by this FISH method, it is likely an underestimate. Quinkert et al. (PMID: 16227280) reported that Con1, which replicates to lower levels than JFH-1, produce ≈40-100 negative-strands per cell and ≈5-8 times more positive strands. More recently, Klepper et al., (PMID: 27642141) found that ≈50% of HCV RNA is double-stranded, which will complicate negative-strand detection. In any case, if the FISH method used here only detects a fraction of negative-strands, it seems likely that it will miss replication compartments that have both R and S genotypes present. Third, it is unclear how representative these micrographs are; please present quantification over a larger number of cells. Fourth, what we'd really like to see is the localization of the R vs. S NS5A proteins in relation to the replication complexes.

6) Figure 4. This figure is quite confusing. First, why use a non-replicating system? Second, why express two NS5As from the same polyprotein? That seems irrelevant in relation to complementation or competition. Third, was there no HA-tagged NS5A expressed in the last two lanes? If so, how was GFP-tagged NS5A co-IP'd?

*Reviewer #3:*

The impaired replication of CA-1 and 2 is very interesting. It might be worth supplying an alignment of these mutant coding sequences in addition to CA-3? Did the first two not replicate because the loss of RNA structures in NS5B?

Figure 3E and F are switched from what is described in the text, which made following these complex concepts more difficult.

In Figure 4D, are the "HA" and "GFP" labels shifted slightly to the right of their intended locations?

Quantify the overlap of RNA strands in Figure 7.

---

## [Author Response]

Essential revisions:1) An experiment at higher MOI to confirm that the cis-dominance and segregation of the replication complexes are not due to asynchronous infection of cells. Even if it is not possible to use a high enough MOI to co-infect every cell, it should be possible to increase the level of co-infection. The authors could then make an argument as to why the MOI is not the cause of the segregation.

A new experiment following this suggestion is shown in Figure 4G. These infections were performed with higher titer stocks so that the abundance of co-infected cells was increased. The duration was also shortened so that only one cycle of infection occurred before the addition of drug. Thus, the synchrony of the infections in the co-infected cells was assured. The same result was obtained, in which a great majority of coinfected cells showed no rescue of drug-susceptible virus. These new data are discussed in the third paragraph of the subsection “Transmission genetics and phenotypic dominance of drug-resistant NS5A variant Y93N”. The increased MOI and 24-hour duration of the experiments is also true for all data now shown in Figure 6.

We also reference two papers that demonstrate the superinfection exclusion discussed by reviewer 1. We consider it likely that, due to this phenomenon, even co-infected cells found at later time points were synchronously infected. This is discussed in the aforementioned paragraph.

2) Further discussion of the literature on complementation between HCV viruses and further discussion of the mechanism of the cis-dominance would help the manuscript.

Thank you for the invitation to discuss more viral genetics! A discussion of the interesting literature of intramolecular and intermolecular proteinase activities, and the genetics of genome rescue and lack thereof, can be found in the third paragraph of the Discussion. We interpret these findings in the context of ours, which we hope can resolve some of the apparent discrepancies between excellent experiments.

Reviewer #1:1) The quality of the scholarship needs improvement. Specifically, the authors didn't cite some relevant publications that do not fit precisely with their model.

We think that the discussion above, in Essential revisions point 2, addresses this point.

2) The use of low MOIs greatly complicates interpretation of the two color FACS data. First, very few cells will be doubly infected during early times of infection at low MOIs; most double positive cells will therefore be the product of asynchronous, secondary infections. Second, because HCV exhibits superinfection exclusion, these co-infections must occur very close in time. The result is that even small differences in fitness may allow one genotype to appear dominant over another simply because it spreads faster and occupies a subset of cells, rather than because of cis dominant drug sensitivity. Thus, replication phenotypes would have to be closely matched for this experimental design to work as reported. This seems most relevant to Figure 6, where the drug-resistant genotype replicated far less efficiently than the wild-type, drug-sensitive genotype (Figure 6C, left panel). If indeed S virus spread more efficiently, it would have prevented coinfection by R virus, sealing its fate. Conversely, if R had replicated more efficiently it would prevented coinfection with S virus, and would not have been burdened by supporting S.

These are good points. The replication efficiencies of the codon-altered and wild-type viruses are well matched, but the numbers of cells infected can differ from experiment to experiment and it is true that small differences could be amplified during multiple cycles. See Essential revisions point 1 above for discussion of new experiments in Figure 4G and Figure 6.

3) Subsection “Transmission genetics and phenotypic dominance of drug-resistant NS5A variant Y93N”, fourth paragraph and Figure 5C. It appears that both S and R NS5A levels were greatly reduced when the S genotype was present during drug treatment. In other words, the expression of the S allele poisoned the expression of the R allele. I recommend that the authors point this out here. Also, this observation complicates the interpretation of Figure 7 that phenotypic mixing does not occur.

The confocal images in Figure 5D indeed make it appear that there is less NS5A protein in the presence of drug in the presence of the drug-susceptible genomes. However, the immunoblots in panels B and C show that the presence of drug or the drug-susceptible genome actually does not affect the abundance of NS5A from either genome. Rather, the dispersed pattern of drug-susceptible NS5A renders the signal less visible than when it is more concentrated.

This has been addressed in the discussion of this figure in the last paragraph of the subsection “Transmission genetics and phenotypic dominance of drug-resistant NS5A variant Y93N”.

4) Subsection “Lack of free mixing may prevent NS5A hetero-oligomerization”, last paragraph and Figure 3—figure supplement 2. The reason why expression of the R allele was unable to rescue the S genome is trivial: DNA transfection renders Huh-7 cells resistant to HCV replication (i.e., the fraction of cells that are productively transfected with DNA are greatly resistant to HCV replication, likely due to induction of an innate antiviral response – go ahead and confirm this with your DNA of choice). Therefore, I expect that very little R was expressed in the same cells as the virus. The obvious approach would be to show a positive control complementation with the same technique used here.

The figure discussed is now Figure 3—figure supplement 2. We agree that a positive control would be excellent but, unfortunately, there are no HCV antivirals that target rescuable products. We are aware of the artifact discussed by the reviewer, and performed the infections three days before the transfections. The control in this experiment is that, in the absence of drug, transfection did not inhibit the growth of virus in the absence of drug (green bars). Nonetheless, it is true that the absence of a positive control makes this experiment incomplete and we would be willing to remove it if requested.

5) Figure 7A. This panel is difficult to interpret for several reasons. First, related to point 2, above, we don't know whether these co-infections occurred simultaneously or sequentially. Second, while we don't know the efficiency of negative- strand detection by this FISH method, it is likely an underestimate. Quinkert et al. (PMID: 16227280) reported that Con1, which replicates to lower levels than JFH-1, produce ≈40-100 negative-strands per cell and ≈5-8 times more positive strands. More recently, Klepper et al., (PMID: 27642141) found that ≈50% of HCV RNA is double-stranded, which will complicate negative-strand detection. In any case, if the FISH method used here only detects a fraction of negative-strands, it seems likely that it will miss replication compartments that have both R and S genotypes present. Third, it is unclear how representative these micrographs are; please present quantification over a larger number of cells. Fourth, what we'd really like to see is the localization of the R vs. S NS5A proteins in relation to the replication complexes.

In response to this point, we have performed a series of new experiments with higher titer viral stocks and 24-hour infections to obtain more synchronized, co-infected cells for purposes of quantitation. Co-localization patterns for S and R negative and positive strands (Figure 6B), S and R negative strands with NS5A, and S and R negative strands with core are shown (Figure 6D). This is discussed in the first two paragraphs of the subsection “Lack of free mixing may prevent NS5A hetero-oligomerization”.

6) Figure 4. This figure is quite confusing. First, why use a non-replicating system? Second, why express two NS5As from the same polyprotein? That seems irrelevant in relation to complementation or competition. Third, was there no HA-tagged NS5A expressed in the last two lanes? If so, how was GFP-tagged NS5A co-IP'd?

Previously submitted Figures 4 and 5 have now been merged into Figure 5 to provide more clarity. Furthermore, the discussion is modified in response to the reviewer’s points in the last two paragraphs of the subsection “Lack of free mixing may prevent NS5A hetero-oligomerization”. The reason to use a non-replicating system and to express the NS5A proteins from the same polyprotein was to ensure biochemical mixing and thus to observe whether it was possible for drug-susceptible NS5A to be dominant when that occurred. There was no HA-tagged NS5A in the rightmost lanes in Figure 5B and 5C because those plasmids contained a single copy of GFP-5A. The lack of GFP-tagged NS5A upon HA immunoprecipitation was included as a control for the specificity of the immunoprecipitation. These experiments are introduced more effectively and clarified in the text in the fourth paragraph of the aforementioned subsection.

Reviewer #3:The impaired replication of CA-1 and 2 is very interesting. It might be worth supplying an alignment of these mutant coding sequences in addition to CA-3? Did the first two not replicate because the loss of RNA structures in NS5B?

Yes, the reviewer is correct that there is an inverse correlation between viability and the presence of recently identified regions of covariation, as mentioned above in response to point 1 of reviewer 2. This is now discussed in the second paragraph of the subsection “Construction of three strains of codon-altered JFH1”. The alignments of all the codon-altered viruses with wild-type JFH1 sequences are now shown in the Figure 1—figure supplements 1-3, and the regions of covariation are noted: two in CA-1, one in CA-2 and none in CA-3.

Figure 3E and F are switched from what is described in the text, which made following these complex concepts more difficult.

Thank you and we apologize for the confusion.

In Figure 4D, are the "HA" and "GFP" labels shifted slightly to the right of their intended locations?

This is remedied in the new figure. Thank you.

Quantify the overlap of RNA strands in Figure 7.

This, and co-localization with NS5A and core, is now shown in Figure 6.